# Upper Limb Changes in DMD Patients Amenable to Skipping Exons 44, 45, 51 and 53: A 24-Month Study

**DOI:** 10.3390/children10040746

**Published:** 2023-04-19

**Authors:** Claudia Brogna, Marika Pane, Giorgia Coratti, Adele D’Amico, Elena Pegoraro, Luca Bello, Valeria Ada Maria Sansone, Emilio Albamonte, Sonia Messina, Antonella Pini, Maria Grazia D’Angelo, Claudio Bruno, Tiziana Mongini, Federica Silvia Ricci, Angela Berardinelli, Roberta Battini, Riccardo Masson, Enrico Silvio Bertini, Luisa Politano, Eugenio Mercuri

**Affiliations:** 1Pediatric Neurology, Università Cattolica del Sacro Cuore, 00168 Rome, Italy; 2Centro Clinico Nemo, Fondazione Policlinico Universitario Agostino Gemelli IRCCS, 00168 Rome, Italy; 3Unit of Neuromuscular and Neurodegenerative Disorders, Bambino Gesù Children’s Hospital, IRCCS, 00165 Rome, Italy; 4Department of Neurosciences, University of Padua, 35128 Padua, Italy; 5The NEMO Center in Milan, Neurorehabilitation Unit, University of Milan, ASST Niguarda Hospital, 20162 Milan, Italy; 6Department of Clinical and Experimental Medicine, University of Messina, 98122 Messina, Italy; 7Neuromuscular Pediatric Unit, IRRCS Istituto delle Scienze Neurologiche di Bologna, 40139 Bologna, Italy; 8NeuroMuscular Unit IRCCS Eugenio Medea, Bosisio Parini, 23842 Lecco, Italy; 9Center of Translational and Experimental Myology and Department of Neuroscience, Rehabilitation, Ophthalmology, Genetics, Maternal and Child Health, IRCCS Istituto Giannina Gaslini and University of Genoa, 16132 Genoa, Italy; 10Neuromuscular Center, AOU Città della Salute e della Scienza, University of Torino, 10100 Turin, Italy; 11National Neurological Institute C. Mondino Foundation, IRCCS, 27100 Pavia, Italy; 12Department of Developmental Neuroscience, IRCCS Stella Maris, 56018 Pisa, Italy; 13Department of Clinical and Experimental Medicine, University of Pisa, 56126 Pisa, Italy; 14Developmental Neurology Unit, Fondazione IRCCS Istituto Neurologico Carlo Besta, 20133 Milan, Italy; 15Cardiomiology and Medical Genetics, Department of Experimental Medicine, Università della Campania Luigi Vanvitelli, 80138 Naples, Italy

**Keywords:** DMD, PUL 2.0, exon skipping

## Abstract

Introduction: The Performance of Upper Limb version 2.0 (PUL 2.0) is increasingly used in Duchenne Muscular Dystrophy (DMD) to study longitudinal functional changes of motor upper limb function in ambulant and non-ambulant patients. The aim of this study was to evaluate changes in upper limb functions in patients carrying mutations amenable to skipping exons 44, 45, 51 and 53. Methods: All DMD patients were assessed using the PUL 2.0 for at least 2 years, focusing on 24-month paired visits in those with mutations eligible for skipping exons 44, 45, 51 and 53. Results: 285 paired assessments were available. The mean total PUL 2.0 12-month change was −0.67 (2.80), −1.15 (3.98), −1.46 (3.37) and −1.95 (4.04) in patients carrying mutations amenable to skipping exon 44, 45, 51 and 53, respectively. The mean total PUL 2.0 24-month change was −1.47 (3.73), −2.78 (5.86), −2.95 (4.56) and −4.53 (6.13) in patients amenable to skipping exon 44, 45, 51 and 53, respectively. The difference in PUL 2.0 mean changes among the type of exon skip class for the total score was not significant at 12 months but was significant at 24 months for the total score (*p* < 0.001), the shoulder (*p* = 0.01) and the elbow domain (*p* < 0.001), with patients amenable to skipping exon 44 having smaller changes compared to those amenable to skipping exon 53. There was no difference within ambulant or non-ambulant cohorts when subdivided by exon skip class for the total and subdomains score (*p* > 0.05). Conclusions: Our results expand the information on upper limb function changes detected by the PUL 2.0 in a relatively large group of DMD patients with distinct exon-skipping classes. This information can be of help when designing clinical trials or in the interpretation of the real world data including non-ambulant patients.

## 1. Introduction

Duchenne muscular dystrophy (DMD) is a X linked disorder with mutations in the dystrophin Xp21 gene resulting in a reduction of dystrophin, a protein that is essential for the stability of the sarcolemma. The diagnosis of DMD is still often achieved after the age of 4 years but the onset of clinical signs is often around the age of 2 to 3 years, at the time when boys fail to achieve the ability to walk fast, run and jump/hop. Historically, DMD boys lose ambulation by the age of 12 years followed by severe respiratory and cardiac impairment leading to death by the age of 18. The introduction of glucocorticoid therapy and improvement in standards of care have changed the natural history of patients affected by DMD, slowing the predictable pattern of progression, improving quality of life and survival.

Over the past two decades several studies have reported natural history data in boys and young adults affected by Duchenne muscular dystrophy [1,2,3,4,5,6]. The availability of new therapeutical approaches specifically targeting subgroups of mutations, such as those eligible for skipping individual exons or those targeting non-sense mutations has shown the need to better understand whether the various genotype subgroups have distinct patterns of progression [7,8,9,10,11,12,13,14]. A few studies have been performed to study longitudinal functional changes in DMD patients with different types of mutations (deletion, duplication, small mutations) and, within the group of deletions, among the subgroups eligible for skipping individual exons, focusing on those skipping 44, 45, 51 and 53 that are the most frequent in DMD boys [2,15,16]. As the clinical trials have been mainly designed for ambulant boys, the great majority of the natural history studies have focused on motor function tests that are relevant for ambulant boys, such as the six minutes walking test (6MWT) and the North Star Ambulatory Assessment (NSAA) [2,15,16,17,18]. Recent studies have reported that there is a difference among the individual subgroups that is not yet significant after 12 months but becomes more marked with increasing follow up [2,18]. Patients carrying mutations amenable to skipping exons 53 and 51 have a lower baseline score on 6MWT and more negative changes than the other subgroups while those with deletions amenable to skipping exon 44 have better baseline scores and less decline [15]. Similar results have been observed when using the NSAA [16].

Less is known about whether differences can also be found in other aspects of motor function, such as upper limb function across the spectrum of functional abilities, including boys who are about to lose ambulation or those who have already lost ambulation, who rely on upper extremities for most of their daily life activities. Recently it has been reported that the Performance of the Upper Limb assessment (PUL) can be reliably used used for longitudinal assessment of upper limb function and its progressive impairment in both ambulant and non-ambulant DMD boys [19,20,21,22]. These studies have shown that, using the PUL, it is possible to follow the overall linear progression from proximal to distal involvement; however, early signs of distal involvement can already be found in relatively young ambulant DMD boys who still have some conserved shoulder function. While a few studies have reported longitudinal upper limb changes in large DMD cohorts, less is known about possible differences among subgroups eligible for skipping individual exons. This has recently become even more important as there is increasing attention and pressure from families and regulators to include non-ambulant older patients in clinical trials.

The aim of our collaborative effort was to obtain longitudinal prospective changes over 24 months in PUL 2.0 in a large cohort of ambulant and non-ambulant DMD patients amenable to skipping exons 44, 45, 51 and 53.

## 2. Material and Methods

### 2.1. Cohort Selection and Dataset Definition

Patients were recruited as part of a larger natural history study. In this study we included all ambulant and non-ambulant patients who had a genetic diagnosis of DMD to establish the overall pattern of 24-month changes in the whole DMD cohort and in the subgroups carrying the three main categories of mutations (deletions, duplications, point mutations). We then focused on more restricted criteria, only including those eligible for skipping 44, 45, 51 and 53. Only patients who had 24-month follow up were included. The study was approved by the institutional review board (ethics committee) of the 13 tertiary participating centers (Catholic University, Rome; Centro Clinico Nemo, University of Milano, Milan; IRCCS Eugenio Medea Bosisio-Parini, Bosisio-Parini; IRCCS Istituto Giannina Gaslini, Genoa; University of Messina, Messina; IRCCS Fondazione Stella Maris and University of Pisa, Pisa; Fondazione IRCCS Istituto Neurologico Besta, Milan; Università della Campania Luigi Vanvitelli, Naples; Ospedale Bambino Gesù, Rome; University of Padua, Padua; Istituto Mondino, Pavia; University of Turin, Turin; Neuromuscular Pediatric Unit, IRCCS Istituto delle Scienze Neurologiche di Bologna, Bologna). Written informed consent was obtained from all participants (or guardians of participants) in the study.

### 2.2. PUL 2.0

The PUL 2.0 includes an entry item to define the starting functional level, and 22 items divided among shoulder level (6 items), middle level (9 items) and distal level (7 items). The maximum score is 42 points (12 for shoulder, 17 for middle level, 13 for distal level).

In previous studies using the NSAA or the 6MWT, the changes became more obvious after 12 months [2,15,16]. In the present study, therefore, we only included patients who had at least 24-month follow-up.

### 2.3. Statistical Analysis

A longitudinal dataset with 24-month paired visits for the same cohort of patients was analyzed to quantify differences in 24-month PUL changes according to the type of mutation and exon-skipping class.

Descriptive statistics were prepared after subdividing the population into ambulatory status classes (ambulant, non-ambulant), type of mutation classes (deletion, duplication, point mutation, other), and exon-skipping classes (44, 45, 51, 53). To provide a descriptive overview of the cohorts, patients who were ambulant at baseline but losing ambulation during the duration of the study were also defined as transitioning patients.

Mean value and standard deviation were reported as descriptive values for the different segments of the population.

Comparisons of difference in PUL 2.0 from the baseline to 12 and/or 24 months were assessed using linear models adjusted with repeated measures.

The ANOVA test with Games–Howell post-hoc test and Bonferroni correction was performed to assess differences in 12 and 24 month changes among exon-skipping classes. Patients with missing data at baseline or 24 months were excluded from the longitudinal analysis, while missing PUL values at 12 months were replaced by linear interpolation. The significance level for statistical tests was set at 0.05. All data processing steps and statistical analyses were performed in SPSS version v27 (BM Corp. Released 2020. IBM SPSS Statistics for Windows, Version 27.0. Armonk, NY, USA: IBM Corp).

## 3. Results

The whole cohort consists of 311 DMD boys (215 with deletions, 26 with duplications and 70 with point mutations) with multiple assessments for a total of 808 paired assessments, 553 from patients with deletions, 60 with duplications and 195 with point mutations.

Within the group of 215 patients with deletions, 27 patients were eligible for skipping exon 44, 25 patients for exon 45, 34 patients for exon 51 and 24 patients for exon 53.

### 3.1. PUL 2.0

In the whole cohort, the PUL 2.0 scores ranged between 0 and 42 at baseline, 12 and 24 months.

When analyzing the paired assessments according to type of mutation, the mean total PUL 2.0 24-month change was −2.90 (5.04) in the whole cohort, −3.02 (4.95) in patients who had deletions, −1.72 (5.09) in patients who had duplications, −3.09 (5.27) in patients who had small mutations.

Changes at 24 months were not statistically significant between type of mutation for the total PUL 2.0 score and for every domain (*p* > 0.05).

### 3.2. Exon Skipping Class

Of the 552 paired assessments included in the deletion’s cohort, 285 (51.63%) were from patients amenable to skipping exon 44, 45, 51 and/or 53. Since patients amenable to skipping exon 51 were, in several cases, amenable to also skipping exon 53, 19 assessments were represented in both skip groups (amenable to skip 51 and 53), for a total number of assessments of 304 (Appendix A).

The mean total PUL 2.0 12-month change was −0.67 (2.80) in patients amenable to skipping exon 44, −1.15 (3.98) in patients amenable to skipping exon 45, −1.46 (3.37) in patients amenable to skipping exon 51 and −1.95 (4.04) in patients amenable to skipping exon 53 (for details see Table 1).

At 12 months, the mean of PUL 2.0 changes was not different among the type of exon- skipping class for the total score (*p* = 0.15), the shoulder (*p* = 0.24), the elbow (*p* = 0.21) or distal domain (*p* = 0.30). There was no difference within ambulant or within non-ambulant patients subdivided by exon-skipping class for the total and subdomains score (*p* > 0.05) (for details, see Appendix A).

The mean total PUL 2.0 24-month change was −1.47 (3.73) in patients amenable to skipping exon 44, −2.78 (5.86) in patients amenable to skipping exon 45, −2.95 (4.56) in patients amenable to skipping exon 51 and −4.53 (6.13) in patients amenable to skipping exon 53.

At 24 months the mean of PUL 2.0 changes was different among the type of exon-skipping class for the total score (*p* < 0.001), the shoulder domain (*p* = 0.01) and the elbow domain (*p* < 0.001), with patients amenable to skipping exon 44 having minor changes if compared to patients amenable to skipping exon 53. The distal domain was not different among the type of exon-skipping class (*p* = 0.50). Details of 24 month-changes are shown in Figure 1 and Table 1. There was no difference within ambulant or non-ambulant patients subdivided by exon-skipping class for the total and subdomains score (*p* > 0.05) (Figure 2).

## 4. Discussion

The PUL 2.0 is increasingly used to study longitudinal changes of functional motor upper limb function in DMD in both clinical and research settings. Because of the rapidly growing number of ongoing or planned trials using the PUL in cohorts eligible for skipping individual exons including non-ambulant patients, there has been increasing interest in defining possible trajectories of PUL 2.0 changes according to exon-skipping classes. To date, very few studies have explored the progression of upper limb function in relation to exon-skipping classes. A large international study focused on ambulatory and non-ambulatory DMD patients eligible for exon 53-skipping used quantitative magnetic resonance imaging and different functional tools (Brooke score, motor function measure, hand grip and key pinch strength, and upper limb distal coordination, MoviPlate) but did not include the PUL [23]. The only study evaluating PUL across exon-skipping classes (version 1.2) reported that patients eligible for skip 53 and skip 51 had lower scores compared to those eligible for skipping exon 44. However, the smaller cohort of 137 participants was probably underpowered to demonstrate significant differences between exon-skipping classes [24].

We analyzed for the first time the 24-month pattern of progression in a cohort of DMD patients amenable to skipping exons 44, 45, 51 and 53.

In previous studies using other functional measures, such as the NSAA or the 6MWT, the changes became more obvious after 12 months; hence, in the present study, we only included patients who had at least 24 months of follow up. The pattern of changes in upper limb functions was similar to that observed in ambulant patients using the 6MWT and NSAA [15,16,25], with individuals eligible for skipping exons 51 and 53 showing a faster progression that those eligible for skipping exon 44. The difference among the individual exon-skipping classes was not significant at 12 months but reached significance at 24 months.

These results are in line with previous observations based on muscle biopsies showing different residual dystrophin expression in the subgroups of deletion of exon 44 that can also be associated with milder phenotypes [25]. Other factors such as modifier genes that were not studied in our cohort may also account for the variability observed [26].

Our results suggest a different progression of changes in patients with distinct genotype classes over 24 months mainly at the shoulder and elbow domains. However, these results should be interpreted with caution. Because of the observational nature of the study, we included all the patients eligible for skipping exon 44, 45, 51 and 53 and a number of variables, such as number of patients, age or baseline PUL, could therefore not be randomised across the individual subgroups.

Another limitation is that the genotype frequencies may not reflect the natural prevalence of the individual subgroups as a number of patients skipping exon 51 and 53 and, to a lesser extent, exon 45, could not be included in the study because they were enrolled in clinical trials.

Despite these limitations, which are shared with all other observational studies reporting natural history data in exon-skipping classes, our findings suggest an effect of the genotype class on progression.

In this paper we also identified some differences between exon-skipping classes from the whole DMD cohort and from the group including all deletions. The recruitment of DMD patients in trials with therapies targeting specific genetic classes is often a major challenge. The possibility of using genotypically matched control groups, especially when selected according to strict age or functional criteria, is limited by the few available longitudinal natural history data. While there is evidence that, especially with increasing follow up, there is a difference in progression among individual exon classes on several measures, including LOA [15,16,17], it has also been reported that the difference between individual exon classes and the mean changes in the whole DMD cohort or in the category with deletions is negligible on a 12-month follow up [18]. A recent study suggested that, at least for a 12-month follow up, genotype class accounted for approximately 2% of variation in one-year changes in the NSAA motor function endpoints [27]. Because of this, a few recent studies have used genotypically unmatched whole DMD cohorts as external comparators, with the suggestion to apply propensity matching or other methods to reduce variability between the samples [28,29].

When we analyzed the differences between the individual exon classes and the total DMD cohort and the subgroup including all deletions, we found that some of the individual exon classes, such as exon 44 and 53, reached statistical significance at 24 months when compared to the mean changes in the whole cohort or in the cohort including all deletions.

In conclusion, our results provide novel information on the upper limb function changes detected by PUL 2.0 in a relatively large group of DMD patients with distinct exon-skipping classes. The patterns of change and the differences observed suggest that this factor should be taken into account when designing clinical trials or in the interpretation of the real-world data also including non-ambulant patients. The data of this study could be useful in establishing the effects of intervention with therapeutical approaches specifically targeting these groups of mutations, such as antisense oligonucleotides. This is particularly important at the present time, since new generations of these drugs are becoming increasingly available in clinical trials.

## Figures and Tables

**Figure 1 children-10-00746-f001:**
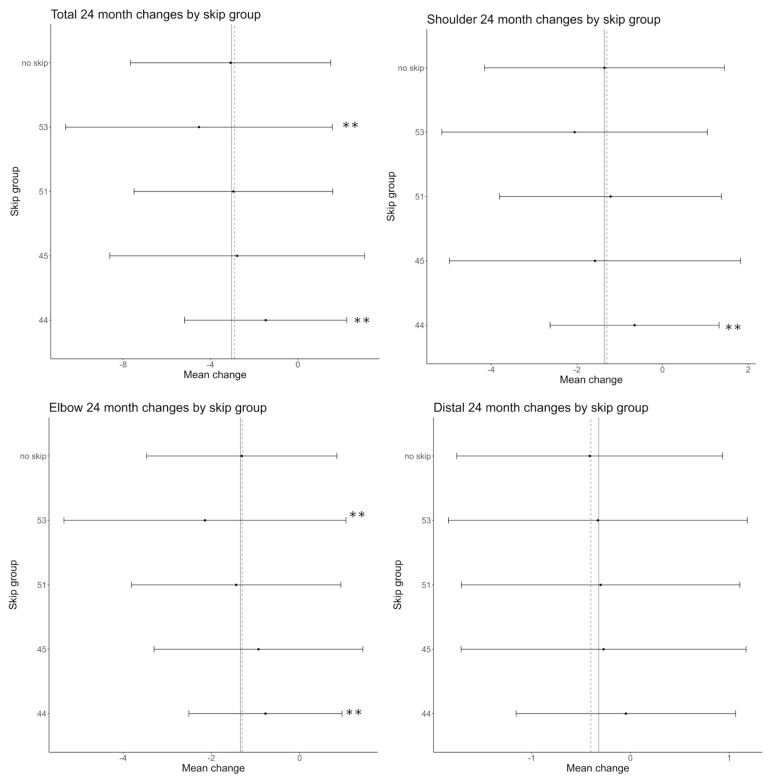
Details of mean changes and SD of the individual exon classes in relation to the mean changes in the whole DMD cohort (vertical dashed line) and in the groups including all deletions (vertical solid line). ** = significant difference from the mean changes of all deletions.

**Figure 2 children-10-00746-f002:**
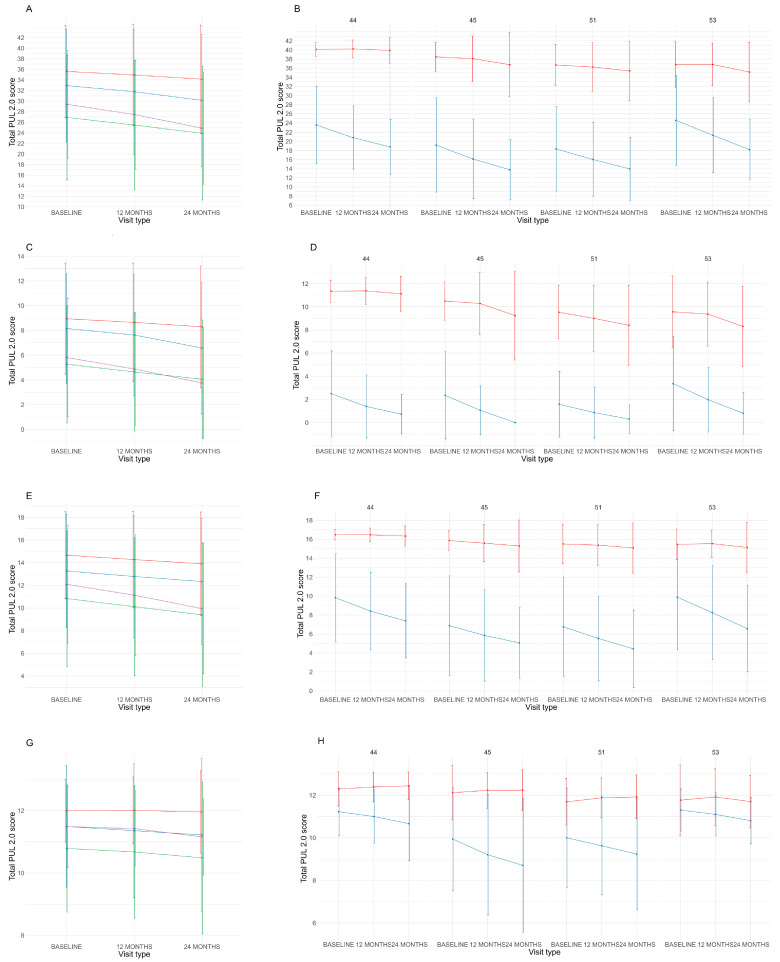
Baseline, 12- and 24-month PUL scores (mean and SD) subdivided in exon-skipping classes (red: amenable to skip 44, blue: amenable to skip 45, green: amenable to skip 51, purple: amenable to skip 53). and to ambulatory status (red: ambulant, blue: non-ambulant). Key to figure: Total score (**A**,**B**), shoulder domain (**C**,**D**), elbow domain (**E**,**F**), distal domain (**G**,**H**).

**Table 1 children-10-00746-t001:** 12- and 24-month PUL changes subdivided in exon skip classes. Key to table: **° =** other deletions not amenable to skipping 44, 45, 51 and 53; ***** = N represent the number of 24-month paired visits.

		44 (N = 66 *)	45 (N = 59 *)	51 (N = 103 *)	53 (N = 76 *)	Other Deletions ° (N = 267 *)
**BASELINE**	**Age**					
	Mean (SD)	12.2 (6.20)	10.3 (4.34)	12.5 (4.96)	13.7 (5.93)	12.6 (5.86)
	Median [Min, Max]	10.2 [3.81, 26.5]	9.49 [3.80, 25.3]	11.2 [4.59, 25.7]	13.0 [4.12, 24.8]	11.4 [3.59, 28.1]
	**TRANSITIONING**					
	NO	62 (93.9%)	54 (91.5%)	86 (83.5%)	62 (81.6%)	236 (88.4%)
	YES	4 (6.1%)	5 (8.5%)	17 (16.5%)	14 (18.4%)	31 (11.6%)
	**TOTAL PUL 2.0**					
	Mean (SD)	35.6 (8.69)	32.9 (10.7)	26.9 (11.8)	29.4 (10.2)	30.3 (12.1)
	Median [Min, Max]	39.5 [8.00, 42.0]	38.0 [3.00, 42.0]	30.0 [3.00, 42.0]	33.0 [10.0, 42.0]	36.0 [0, 42.0]
**12 MONTH changes**	**TOTAL PUL 2.0**					
	Mean (SD)	−0.674 (2.80)	−1.15 (3.98)	−1.46 (3.37)	−1.95 (4.04)	−1.39 (3.19)
	Median [Min, Max]	0 [−12.0, 5.00]	0 [−17.0, 11.0]	−1.00 [−13.0, 8.00]	−2.00 [−19.0, 8.00]	−1.00 [−14.0, 7.00]
	**SHOULDER PUL 2.0**					
	Mean (SD)	−0.288 (1.54)	−0.525 (2.26)	−0.626 (1.84)	−0.921 (2.16)	−0.569 (1.99)
	Median [Min, Max]	0 [−5.00, 3.00]	0 [−8.00, 6.00]	0 [−6.00, 4.00]	0 [−9.00, 6.00]	0 [−10.0, 6.00]
	**ELBOW PUL 2.0**					
	Mean (SD)	−0.394 (1.27)	−0.492 (1.76)	−0.718 (1.84)	−0.961 (2.10)	−0.710 (1.68)
	Median [Min, Max]	0 [−6.00, 1.00]	0 [−8.00, 4.00]	0 [−7.00, 5.00]	0 [−10.0, 3.00]	0 [−7.00, 3.00]
	**DISTAL PUL 2.0**					
	Mean (SD)	0.00758 (0.825)	−0.136 (1.18)	−0.112 (1.08)	−0.0658 (1.06)	−0.114 (1.03)
	Median [Min, Max]	0 [−4.00, 2.00]	0 [−3.00, 5.00]	0 [−3.00, 3.00]	0 [−4.00, 3.00]	0 [−4.00, 3.00]
**24 MONTH changes**	**TOTAL PUL 2.0**					
	Mean (SD)	−1.47 (3.73)	−2.78 (5.86)	−2.95 (4.56)	−4.53 (6.13)	−3.08 (4.59)
	Median [Min, Max]	0 [−13.0, 5.00]	−1.00 [−18.0, 11.0]	−3.00 [−16.0, 8.00]	−4.00 [−25.0, 11.0]	−2.00 [−18.0, 9.00]
	**SHOULDER PUL 2.0**					
	Mean (SD)	−0.652 (1.98)	−1.58 (3.41)	−1.21 (2.60)	−2.05 (3.11)	−1.36 (2.80)
	Median [Min, Max]	0 [−8.00, 3.00]	0 [−9.00, 6.00]	0 [−9.00, 5.00]	0 [−11.0, 4.00]	0 [−12.0, 5.00]
	**ELBOW PUL 2.0**					
	Mean (SD)	−0.773 (1.74)	−0.932 (2.38)	−1.44 (2.38)	−2.14 (3.21)	−1.31 (2.16)
	Median [Min, Max]	0 [−6.00, 1.00]	0 [−8.00, 4.00]	−1.00 [−8.00, 4.00]	−2.00 [−12.0, 6.00]	0 [−10.0, 3.00]
	**DISTAL PUL 2.0**					
	Mean (SD)	−0.0455 (1.12)	−0.271 (1.45)	−0.301 (1.41)	−0.329 (1.52)	−0.412 (1.35)
	Median [Min, Max]	0 [−6.00, 2.00]	0 [−4.00, 5.00]	0 [−4.00, 4.00]	0 [−6.00, 6.00]	0 [−6.00, 4.00]

## Data Availability

All data is contained within the article.

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
