# Peer review of "Upper Limb Changes in DMD Patients Amenable to Skipping Exons 44, 45, 51 and 53: A 24-Month Study"

_children, 2023, doi:10.3390/children10040746_

Round 1

Reviewer 1 Report

The NSAA is a 17-item scale that grades the performance of various functional skills on a scale from 0-2 and needs multiple assessments.

Performance of upper limb version 2.0 (PUL 2.0) is more simple than the NSAA test.

In the current manuscript, PUL 2.0 was used to assess the natural history for DMD patients amenable to exon skipping 44, 45, 51, and 53. Since exon-skipping therapy becomes available for exon 53, and therapies against other exons are in clinical studies, the manuscript may be useful for forthcoming exon-skipping therapies.

The comments are as follows.

The data showing patients amenable to exon 44 skipping had smaller changes compared with others including exon 53 is informative. It provides important information. However, the authors already published similar data using NSAA in 2021, and 36 month studies were reported for DMD in multiple countries. This manuscript is a study for 24 month only in Italy. Therefore, the limitation and the merit of the current study should be more clearly discussed.

Lines 39 and 162, are -2.95 (4.54) correct? In Table 1, it is written as -2.95 (4.56).

In Table 1, it could be better to write exon 44(n=66 A) and so on. What does A in (n=66 A) mean? Is it for ambulant?

In Figure 1 legend, ** is not found in the actual figure, I assume. First of all, Figures 1 and 2 are hard to see, possibly due to the overlap of colors, and small letters.

The number of patients is not easy to follow.

The reviewer understands that of the 552 paired assessments, 285 were selected from the patients amenable to exon skipping 44, 45, 51 and/or 53. However, in Table 1, n=267. How does this number come from? In line 149, it is described that 304 patients were selected. Does this number appear somewhere in the text?

Author Response

The NSAA is a 17-item scale that grades the performance of various functional skills on a scale from 0-2 and needs multiple assessments.

Performance of upper limb version 2.0 (PUL 2.0) is more simple than the NSAA test.

In the current manuscript, PUL 2.0 was used to assess the natural history for DMD patients amenable to exon skipping 44, 45, 51, and 53. Since exon-skipping therapy becomes available for exon 53, and therapies against other exons are in clinical studies, the manuscript may be useful for forthcoming exon-skipping therapies.

The comments are as follows.

The data showing patients amenable to exon 44 skipping had smaller changes compared with others including exon 53 is informative. It provides important information. However, the authors already published similar data using NSAA in 2021, and 36 month studies were reported for DMD in multiple countries. This manuscript is a study for 24 month only in Italy. Therefore, the limitation and the merit of the current study should be more clearly discussed.

We thank the reviewer for this comment. Unfortunately, we could not use data from other countries as they implemented the PUL 2.0 version at a later stage compared to us and 24-month data is not available.

Lines 39 and 162, are -2.95 (4.54) correct? In Table 1, it is written as -2.95 (4.56).

We thank the reviewer for highlight this, we confirm that was a typo and the text has been amended.

In Table 1, it could be better to write exon 44(n=66 A) and so on. What does A in (n=66 A) mean? Is it for ambulant?

We agree with the reviewer that “A” was confusing. We have modified the symbols and provided explanations for the reader in the “key to the table” legend.

In Figure 1 legend, ** is not found in the actual figure, I assume. First of all, Figures 1 and 2 are hard to see, possibly due to the overlap of colors, and small letters.

We thank the reviewer for highlight this, we have amended the figure 1 and tried to magnify figure 2.

 The number of patients is not easy to follow.

The reviewer understands that of the 552 paired assessments, 285 were selected from the patients amenable to exon skipping 44, 45, 51 and/or 53. However, in Table 1, n=267. How does this number come from? In line 149, it is described that 304 patients were selected. Does this number appear somewhere in the text?

We agree with the reviewer that this was probably hard to follow. Regarding table 1, we have now better highlighted that N=267 was referring to “other deletions” not amenable to skip exons 44,45,52,53. Regarding line 149, since patients amenable to skip exon 51 were, in several cases, amenable to also skip exon 53, 19 assessments were represented in both skip groups (amenable to skip 51 and 53), for a total number of assessments of 304. This has been also added to the main text.

Reviewer 2 Report

The authors have initiated a detailed study of changes in upper limb function in individuals with Duchenne muscular dystrophy.

By characterising trends in cohorts of patients stratified on the basis of their dystrophin mutation (and amenability to exon skipping) should contribute to a more solid base-line from which to assess functional changes in response to the exon skipping therapy.

From this work, I believe there are a couple of very important points that perhaps could be emphasised in the manuscript and/or highlighted in the conclusion.

The ability to assess upper limb function will allow non-ambulant individuals to be included in any of the future studies, an important feature to assess as there should be a substantial number of non-ambulant patients receiving exon skipping therapies (or even gene replacemnet therapies).

Another important point is that while DMD is a relentlessly progressive muscle wasting disease, the deterioration is slow.  While changes were not obvious at 12 months, the differences were significant at 2 years.  As I understand the disease progression, over the age or 7 or 8 years, there is only slow deterioration, until ambulation is lost.

There are subtle differences in disease severity between the groups amenabale to different exon skipping treatments, with exon 44 skipping having smaller changes than those who may benefit from skipping exon 53. 

Could the authors speculate as to why this is so?  Some 3 decades ago, Louise Nicholson and colleagues reported that boys with no detectable dystrophin (by western blotting) lost ambulation some 2 years before DMD boys who showed a trace of dystrophin. (this was the pre-steroid era).

Could it be that the exon 44/45 amenable patients are making higher levels of this trace dystrophin through naturally occurring errors in splicing associated with the massive intron 44 (~248 kB).  Parts of the dystrophin gene have been described as having leaky exppression with alternatively exon skipped transcripts more frequently detected.

Author Response

This is a well-written manuscript on the physical therapy examination of DMD patients. The authors reported a different progression of changes in PUL 2.0 in DMD patients with distinct genotype classes over 24 months. Although the authors admitted that their results should be interpreted with caution (line 198), the manuscript suggested the effectiveness of exon-skipping therapy may be dependent on the genotype classes, even if the best anti-sense oligonucleotide drugs are synthesized for each target exon.  

I completely agree with the authors’ statement that their information can be of help when designing clinical trials or in the interpretation of the real-world data (Abstract and Discussion).

However, I have some questions about this manuscript, which will be shown as “major issues” and “minor issues”.

[Major issues of the manuscript]

(1)    About sentences at lines 156-160  “At 12 months the mean of PUL 2.0 changes was not different among the type of exon skip class for the total score (p = 0.15), the shoulder (p = 0.24), the elbow (p = 0.21) or distal domain (p = 0.30). There was no difference within ambulant or non-ambulant patients subdivided by exon skip class for the total and subdomains score (p > 0.05) (for details see Table 1).” My questions are as follows;

①    Patients in Table 1 are subdivided into exon skip classes, but not ambulant or non-ambulant patient groups. Thus, I cannot check whether there was no difference within ambulant or non-ambulant patients subdivided by exon skip class for the total and subdomains score.

②     If the authors want to say that the mean of PUL 2.0 changes was not different between ambulant or non-ambulant groups in the same exon skip class, it is necessary to show the data of ambulant or non-ambulant patients for the total and subdomains score in each exon skip class.

③     However, predicting from the data of Figure 1, there might be some differences between ambulant or non-ambulant groups in the same exon skip class. Please look at the patients with “DELETION” in Figure 1. We can see smaller changes in ambulant patients compared to non-ambulant patients for the total and subdomains score. When the ambulant patient number is so small in this study, the authors may not find the statistically significant difference.

 For all 3 comments: This can be checked from figure 2, in which mean and SD are reported for baseline, 12 and 24 months PUL scores subdivided in exon skip classes and to ambulatory status. If the reviewer and the editor feel that we should duplicate this information, we can add It also to the table or as supplementary file.

[Minor issues of the manuscript]

(1)    About Table 1  My questions are as follows;

①     “Transitioning” is used only in Table 1, and there are no explanations of this word in the manuscript. I am wondering whether this word is related with “ambulant”. If it is related with “ambulant”, I think “transitioning” should be replaced by “ambulant”.

We agree with the reviewer that this information was missing, we have added the definition in the methodology section (lines 118-120).

②     I am wondering why the authors used assessment numbers in this study. To avoid the misunderstanding of the readers, patient numbers, participant numbers and assessment numbers in each genotype should be shown in an additional table.

The information about number of patient per each genotype was present in lines 132-135 and per exon skip class in lines 136-137, the information about number of assessments was indeed present in in lines 132-135 and in table 1. We have now provided a supplementary table to summarize this data.

Reviewer 3 Report

[Summary of the manuscript]

In this manuscript entitled “Upper limb changes in DMD patients amenable to skip exons 2 44, 45, 51 and 53: a 24 month study”, the authors used the Performance of upper limb version 2.0 (PUL 2.0) in Duchenne muscular Dystrophy (DMD) to study longitudinal functional changes of motor upper limb function, focusing on the 24-month paired visits in the patients with mutations eligible for skipping exons 44, 45, 51 and 53. The mean total PUL 2.0 24-month change was -1.47 (3.73), -2.78 (5.86), -2.95 (4.54) and -4.53 (6.13) in patients 39 amenable to skip exon 44, 45, 51 and 53 respectively. The difference in PUL 2.0 mean changes among the type of exon skip class for the total score was significant at 24 months for the total score (p < 0.001), the shoulder (p = 0.01) and the elbow domain (p < 0.001) with patients amenable to skip exon 44 having smaller changes compared to those amenable to skip exon 53. According to their results, the upper limb function changes detected by the PUL 2.0 varied in DMD patients with distinct exon skipping classes. The authors concluded that such information can be of help when designing clinical trials or in the interpretation of the real-world data.

[General comments to the manuscript]

This is a well-written manuscript on the physical therapy examination of DMD patients. The authors reported a different progression of changes in PUL 2.0 in DMD patients with distinct genotype classes over 24 months. Although the authors admitted that their results should be interpreted with caution (line 198), the manuscript suggested the effectiveness of exon-skipping therapy may be dependent on the genotype classes, even if the best anti-sense oligonucleotide drugs are synthesized for each target exon.  

I completely agree with the authors’ statement that their information can be of help when designing clinical trials or in the interpretation of the real-world data (Abstract and Discussion).

However, I have some questions about this manuscript, which will be shown as “major issues” and “minor issues”.

[Major issues of the manuscript]

(1)    About sentences at lines 156-160  “At 12 months the mean of PUL 2.0 changes was not different among the type of exon skip class for the total score (p = 0.15), the shoulder (p = 0.24), the elbow (p = 0.21) or distal domain (p = 0.30). There was no difference within ambulant or non-ambulant patients subdivided by exon skip class for the total and subdomains score (p > 0.05) (for details see Table 1).” My questions are as follows;

     Patients in Table 1 are subdivided into exon skip classes, but not ambulant or non-ambulant patient groups. Thus, I cannot check whether there was no difference within ambulant or non-ambulant patients subdivided by exon skip class for the total and subdomains score.

     If the authors want to say that the mean of PUL 2.0 changes was not different between ambulant or non-ambulant groups in the same exon skip class, it is necessary to show the data of ambulant or non-ambulant patients for the total and subdomains score in each exon skip class.

     However, predicting from the data of Figure 1, there might be some differences between ambulant or non-ambulant groups in the same exon skip class. Please look at the patients with “DELETION” in Figure 1. We can see smaller changes in ambulant patients compared to non-ambulant patients for the total and subdomains score. When the ambulant patient number is so small in this study, the authors may not find the statistically significant difference.

[Minor issues of the manuscript]

(1)    About Table 1  My questions are as follows;

     “Transitioning” is used only in Table 1, and there are no explanations of this word in the manuscript. I am wondering whether this word is related with “ambulant”. If it is related with “ambulant”, I think “transitioning” should be replaced by “ambulant”.

     I am wondering why the authors used assessment numbers in this study. To avoid the misunderstanding of the readers, patient numbers, participant numbers and assessment numbers in each genotype should be shown in an additional table.

Author Response

The authors have initiated a detailed study of changes in upper limb function in individuals with Duchenne muscular dystrophy.

By characterising trends in cohorts of patients stratified on the basis of their dystrophin mutation (and amenability to exon skipping) should contribute to a more solid base-line from which to assess functional changes in response to the exon skipping therapy.

From this work, I believe there are a couple of very important points that perhaps could be emphasised in the manuscript and/or highlighted in the conclusion.

The ability to assess upper limb function will allow non-ambulant individuals to be included in any of the future studies, an important feature to assess as there should be a substantial number of non-ambulant patients receiving exon skipping therapies (or even gene replacemnet therapies).

Another important point is that while DMD is a relentlessly progressive muscle wasting disease, the deterioration is slow.  While changes were not obvious at 12 months, the differences were significant at 2 years.  As I understand the disease progression, over the age or 7 or 8 years, there is only slow deterioration, until ambulation is lost.

There are subtle differences in disease severity between the groups amenabale to different exon skipping treatments, with exon 44 skipping having smaller changes than those who may benefit from skipping exon 53. 

Could the authors speculate as to why this is so?  Some 3 decades ago, Louise Nicholson and colleagues reported that boys with no detectable dystrophin (by western blotting) lost ambulation some 2 years before DMD boys who showed a trace of dystrophin. (this was the pre-steroid era).

Could it be that the exon 44/45 amenable patients are making higher levels of this trace dystrophin through naturally occurring errors in splicing associated with the massive intron 44 (~248 kB).  Parts of the dystrophin gene have been described as having leaky exppression with alternatively exon skipped transcripts more frequently detected.

We thank the reviewer for the comments.

These results are in line with previous observations based on muscle biopsies showing different residual dystrophin expression in the subgroups of deletion of exon 44 that can also be associated with milder phenotypes [Anthony K, Arechavala-Gomeza V, Ricotti V, Torelli S, Feng L, Janghra N, et al. Biochemical characterization of patients with in-frame or out-of-frame DMD deletions pertinent to exon 44 or 45 skipping. JAMA Neurol. 2014; 71(1):32–4; ];

Other factors such as modifier genes [Bello L, Kesari A, Gordish-Dressman H, Cnaan A, Morgenroth LP, Punetha J, et al. Genetic modifiers of ambulation in the Cooperative International Neuromuscular Research Group Duchenne Natural History Study. Ann Neurol. 2015; 77(4):684–696 ] that were not studied in our cohort may also account for the variability observed

These considerations have been added in the discussion and the references also have been added.

Round 2

Reviewer 3 Report

(1)   About Table 1 and Figure 2

My previous comment (1): Patients in Table 1 are subdivided into exon skip classes, but not ambulant or non-ambulant patient groups. Thus, I cannot check whether there was no difference within ambulant or non-ambulant patients subdivided by exon skip class for the total and subdomains score.

The authors’ reply: “This can be checked from figure 2”

My new comment to the reply: Regarding the description “there was no difference within ambulant or non-ambulant patients” (lines 158-159 of the original version), I wrote the comment shown above in my first review report.

If my understanding is correct, the data of Table 1 may not distinguish ambulant or non-ambulant patients. But the authors placed the note (for details see Table 1) just after the sentence “there was no difference within ambulant or non-ambulant patients”. The note led me to look for such information in Table 1.

In the revised version, the index paragraph including the note has not been changed. I think it may be better to revise the paragraph including the note.

(2) About detailed data of Figure 1

My previous comment (1): If the authors want to say that the mean of PUL 2.0 changes was not different between ambulant or non-ambulant groups in the same exon skip class, it is necessary to show the data of ambulant or non-ambulant patients for the total and subdomains score in each exon skip class. However, predicting from the data of Figure 1, there might be some differences between ambulant or non-ambulant groups in the same exon skip class. Please look at the patients with “DELETION” in Figure 1. We can see smaller changes in ambulant patients compared to non-ambulant patients for the total and subdomains score. When the ambulant patient number is so small in this study, the authors may not find the statistically significant difference.

The authors’ reply: “If the reviewer and the editor feel that we should duplicate this information, we can add It also to the table or as supplementary file.”

My new comment to the reply: I am happy to hear that the authors are ready to present the detailed data as supplementary data.

Researchers in this field may want to know the initiation timing and duration period of antisense treatment for DMD patients. I believe the data of the authors in this study will be very useful in establishing the treatment algorithm. That is why I stick to the status of ambulant or non-ambulant patients.

(3)   About three groups (ambulant in the study period, transitioning, and non-ambulant in the study period)

My previous comment (2): “Transitioning” is used only in Table 1, and there are no explanations of this word in the manuscript. I am wondering whether this word is related with “ambulant”. If it is related with “ambulant”, I think “transitioning” should be replaced by “ambulant”.

The authors’ reply: “We agree with the reviewer that this information was missing, we have added the definition in the methodology section (lines 118-120).”

Revision (at lines 118-120 of the revised version) Patients who were ambulant at baseline but losing ambulation during the duration of the study were also defined as transitioning patients.

My new comment to the reply: Based on the authors’ explanation, the patients in this study should be divided into three groups; ambulant during the study period, transitioning, and non-ambulant during the study period. 

However, only two groups are shown in Table 1: transitioning and non-transitioning. Then we cannot distinguish “ambulant during the study period” patients and “non-ambulant during the study period” patients, because they may belong to non-transitioning group.

In addition, Figure 2 showed the data based on the classification of ambulant and non-ambulant patients. I cannot see how the authors handled transitioning patients in Figure 2.

To show the consistency between Table 1 and Figure 2, it may be necessary to remake Figure 2, according to the classification of patients who were ambulant in the study period, transitioning, and non-ambulant in the study period.

(4) About necessity of showing “No bias towards any particular patient and timing”

My previous comment (2): I am wondering why the authors used assessment numbers in this study. To avoid the misunderstanding of the readers, patient numbers, participant numbers and assessment numbers in each genotype should be shown in an additional table.

The authors’ reply: “The information about number of patient per each genotype was present in lines 132-135 and per exon skip class in lines 136-137, the information about number of assessments was indeed present in in lines 132-135 and in table 1. We have now provided a supplementary table to summarize this data.”

My new comment to the reply: I appreciate the authors’ effort providing a supplementary table. However, the table should also contain some other information (for example, the number of patients who were checked at baseline, 12 months and 24 months). The table is not enough to guarantee that the data is not arbitrary, or the assessments are not biased towards any particular patient and timing. I ask the authors to add some description which guarantee no bias towards any particular patient and timing

(5) Summary

The authors presented important data for the treatment for DMD patients. As mentioned above, I believe the data of this study will be very useful in establishing the antisense treatment algorithm. That is why I ask the authors to show a more complete description of their data.

Author Response

(1)   About Table 1 and Figure 2

My previous comment (1)①: Patients in Table 1 are subdivided into exon skip classes, but not ambulant or non-ambulant patient groups. Thus, I cannot check whether there was no difference within ambulant or non-ambulant patients subdivided by exon skip class for the total and subdomains score.

 The authors’ reply: “This can be checked from figure 2”

 My new comment to the reply: Regarding the description “there was no difference within ambulant or non-ambulant patients” (lines 158-159 of the original version), I wrote the comment shown above in my first review report.

 If my understanding is correct, the data of Table 1 may not distinguish ambulant or non-ambulant patients. But the authors placed the note (for details see Table 1) just after the sentence “there was no difference within ambulant or non-ambulant patients”. The note led me to look for such information in Table 1.

 In the revised version, the index paragraph including the note has not been changed. I think it may be better to revise the paragraph including the note

In the light of reviewer comment 1 and 2 we’ve now added a supplementary table to the text. Please see supplementary table 2.

(2) About detailed data of Figure 1

My previous comment (1)②: If the authors want to say that the mean of PUL 2.0 changes was not different between ambulant or non-ambulant groups in the same exon skip class, it is necessary to show the data of ambulant or non-ambulant patients for the total and subdomains score in each exon skip class. ③ However, predicting from the data of Figure 1, there might be some differences between ambulant or non-ambulant groups in the same exon skip class. Please look at the patients with “DELETION” in Figure 1. We can see smaller changes in ambulant patients compared to non-ambulant patients for the total and subdomains score. When the ambulant patient number is so small in this study, the authors may not find the statistically significant difference.

 The authors’ reply: “If the reviewer and the editor feel that we should duplicate this information, we can add It also to the table or as supplementary file.”

 My new comment to the reply: I am happy to hear that the authors are ready to present the detailed data as supplementary data.

 Researchers in this field may want to know the initiation timing and duration period of antisense treatment for DMD patients. I believe the data of the authors in this study will be very useful in establishing the treatment algorithm. That is why I stick to the status of ambulant or non-ambulant patients.

 In the light of reviewer comment 1 and 2 we’ve now added a supplementary table to the text. Please see supplementary table 2.

(3)   About three groups (ambulant in the study period, transitioning, and non-ambulant in the study period)

My previous comment (2)①: “Transitioning” is used only in Table 1, and there are no explanations of this word in the manuscript. I am wondering whether this word is related with “ambulant”. If it is related with “ambulant”, I think “transitioning” should be replaced by “ambulant”.

The authors’ reply: “We agree with the reviewer that this information was missing, we have added the definition in the methodology section (lines 118-120).”

Revision (at lines 118-120 of the revised version) Patients who were ambulant at baseline but losing ambulation during the duration of the study were also defined as transitioning patients.

 My new comment to the reply: Based on the authors’ explanation, the patients in this study should be divided into three groups; ambulant during the study period, transitioning, and non-ambulant during the study period. 

However, only two groups are shown in Table 1: transitioning and non-transitioning. Then we cannot distinguish “ambulant during the study period” patients and “non-ambulant during the study period” patients, because they may belong to non-transitioning group.

In addition, Figure 2 showed the data based on the classification of ambulant and non-ambulant patients. I cannot see how the authors handled transitioning patients in Figure 2.

 To show the consistency between Table 1 and Figure 2, it may be necessary to remake Figure 2, according to the classification of patients who were ambulant in the study period, transitioning, and non-ambulant in the study period.

The aim of this study was not to assess differences between ambulant in the study period, transitioning, and non-ambulant in the study period. Furthermore, the study sample size does not allow for any statistical analysis with three subgroups. We have reported this information in table 1 as % of patients who where transitioning during the study period so to contextualize the results for the reader and provide a descriptive overview of the cohort. We have added this specification in the methods section (lines 124-126).

(4) About necessity of showing “No bias towards any particular patient and timing”

My previous comment (2)②: I am wondering why the authors used assessment numbers in this study. To avoid the misunderstanding of the readers, patient numbers, participant numbers and assessment numbers in each genotype should be shown in an additional table.

 The authors’ reply: “The information about number of patient per each genotype was present in lines 132-135 and per exon skip class in lines 136-137, the information about number of assessments was indeed present in in lines 132-135 and in table 1. We have now provided a supplementary table to summarize this data.”

My new comment to the reply: I appreciate the authors’ effort providing a supplementary table. However, the table should also contain some other information (for example, the number of patients who were checked at baseline, 12 months and 24 months). The table is not enough to guarantee that the data is not arbitrary, or the assessments are not biased towards any particular patient and timing. I ask the authors to add some description which guarantee no bias towards any particular patient and timing.

As stated in the methods section “Patients with missing data at baseline or 24 months were excluded from the longitudinal analysis” Therefore, each patient has 24 months of follow-up and no other selection has been made. We have added this to the key to table legend of supplementary table 1.

(5) Summary

The authors presented important data for the treatment for DMD patients. As mentioned above, I believe the data of this study will be very useful in establishing the antisense treatment algorithm. That is why I ask the authors to show a more complete description of their data.

Additional data, as suggested by the reviewer, are now showed in the supplementary tables.